# Prevalence and risk factors of geohelminthiasis among the rural village children in Kota Marudu, Sabah, Malaysia

**A. Lim-Leroy, Tock H. Chua**⬥*

Department of Pathobiology and Medical Diagnostics, Faculty of Medicine and Health Sciences, Universiti Malaysia Sabah, Kota Kinabalu, Malaysia

* thchua@ums.edu.my

**Data Availability Statement:** All relevant data are within the manuscript and its Supporting Information files.

**Funding:** ALL was partly supported by United States Agency for International Development

## Abstract

Geohelminthiasis is a worldwide problem, especially in low-income countries. Children from rural areas and those living in poverty, lacking basic health amenities and having poor environmental sanitation are likely to be affected. Adverse effects such as anemia, protein malnutrition, colitis are common which can affect both the children's physical and mental growing development. A cross-sectional study on geohelminthiasis was conducted among children from 238 households in 13 villages in Kota Marudu of northern Sabah, East Malaysia. The study involved interviewing villagers using questionnaires to collect demographic and socio-economic data, getting faecal samples from the children, collecting soil samples and identifying parasite eggs with microscopy and molecular methods. A total of 407 children (6 months-17 years old) enrolled in the study. Geohelminthiasis was detected in the faecal samples of children from 54% (7/13) of the villages with mean prevalence of infection per village of 9.0% (0%-34.9%). On a household basis, 18% (43/238) of the households sampled had infected children, with mean prevalence rate per household of 11% (0%-43%). The prevalence was for *Ascaris lumbricoides*: 9.6% (39/407), *Trichuris trichiura*: 2.7% (11/407) and hookworms (*Necator americanus* and *Ancylostoma sp.*): 2.7% (11/407). The overall mean infection rate of the children examined was 14.3%. Significantly higher prevalence was recorded for the children of mothers who did not have any formal education (p = 0.003); household income of less than USD119 (RM500) (p<0.001); children from homes without proper sanitation facilities (p<0.001); children who usually go about barefoot (p<0.001) and not washing feet before entering the house (p = 0.017). Soil samples were found to have geohelminth eggs or larvae which could be due to unhygienic sanitation practices. This study shows the geohelminthiasis is prevalent in the villages, and the risk factors are lack of maternal education, low income, poor sanitation facilities and irregular deworming practice. Expanding deworming coverage in the study region may help reduce the worm infections in these communities, so that the mental and physical development of the children would not be affected by geohelminthiasis. The data on the prevalence of geohelminthiasis in this study would contribute to better public health monitoring and operation to reduce the infection in rural areas.

(USAID) Infectious Disease Emergence and Economics of Altered Landscapes: (IDEEAL) project, (Cooperative Agreement number AID-486-A-13-00005), and partly supported by Malaysian Ministry of Education research grant GSP001 awarded to THC. There was no additional external funding received for this study.

**Competing interests:** The authors have declared that no competing interests exist.

## Introduction

The term soil-transmitted helminths (STH) or geohelminths refers to *Ascaris lumbricoides*, hookworms or *Trichuris trichiura*, and infection by these helminths is collectively known as geohelminthiasis. It is a worldwide problem, especially in low-income countries.

In 2010, it was estimated that globally 438.9 million people were infected with hookworm, 819.0 million with *A. lumbricoides* and 464.6 million with *T. trichiura* [1]. Chronic infections of *Ascaris*, *Trichuris*, and hookworm in children have adverse effects on their physical and mental development, and the severity of these effects are dependent on the intensity of the infection [2].

Children, especially from rural areas, living in poverty, lacking basic amenities and having poor environmental sanitation and hygiene, are often infected with geohelminths. Infection by *Ascaris lumbricoides* rarely causes death, but it can cause anemia affecting children's physical and mental growing development [3, 4]. Heavy hookworm infection in children could lead to "hookworm disease" [5] with characteristic iron deficiency anemia and protein malnutrition resulting from intestinal blood loss. More importantly, anemia during pregnancy can result in prematurity, low birthweight, and impaired lactation [6]. Infection by *Trichuris trichiura* is comparatively less harmful and without symptoms although medium to heavy infections can lead to colitis, with associated chronic abdominal pain and diarrhoea, and *Trichuris* dysentery syndrome [7]. In Malaysia, geohelminthiasis is considered endemic, with the most common STH being *Ascaris lumbricoides*, *Trichuris trichiura* and hookworm [8]. Although the living standards of the Malaysian rural communities and indigenous people have been upgraded with better facilities through the government's efforts, high incidences of geohelminthiasis are still being reported [9, 10].

Several studies of geohelminth infection have been conducted in West Malaysia among the children of various communities, backgrounds and settings. High prevalence of *A. lumbricoides* (59.9%) and *T. trichiura* (47.1%) were detected in the children of the indigenous communities (*Orang Asli*) [11]. Sinniah et al. [12] reviewed 101 studies covering 42 years of research on intestinal parasitic infections among communities living in different habitats and recorded high incidence of STH infection among the indigenous children. Globally, it appears that Malaysia had the highest prevalence of *T. trichiura* infections (49.9%) [1].

Comparatively fewer studies had been conducted in the rural areas of both states of Sabah and Sarawak of East Malaysia [13, 14] which have provided some basic information on the prevalence of STH infection. This study was conducted at 13 rural communities in the district of Kota Marudu, Sabah, to determine the prevalence of geohelminthiasis, particularly ascariasis among the indigenous children of these villages between the age of 6-month-old to 17-years as well as to identify the potential associated risk factors.

## Materials and methods

This study was approved by the Malaysian National Medical Research Register (NMRR, Ref. 25014_20161014_070033_067). Signed consent was obtained from the village heads of all villages, and the parents of the children before data collection commenced.

### Study area and population

The study, lasting from April 2015 to September 2017, was carried out at Kota Marudu district, located in the northern region and which is one of the districts in Sabah currently undergoing rapid urbanization (Fig 1).

Thirteen villages in Kota Marudu district (out of 138) were selected as study sites, based on the following criteria: they must be rural, more than 5 km away from Kota Marudu town,

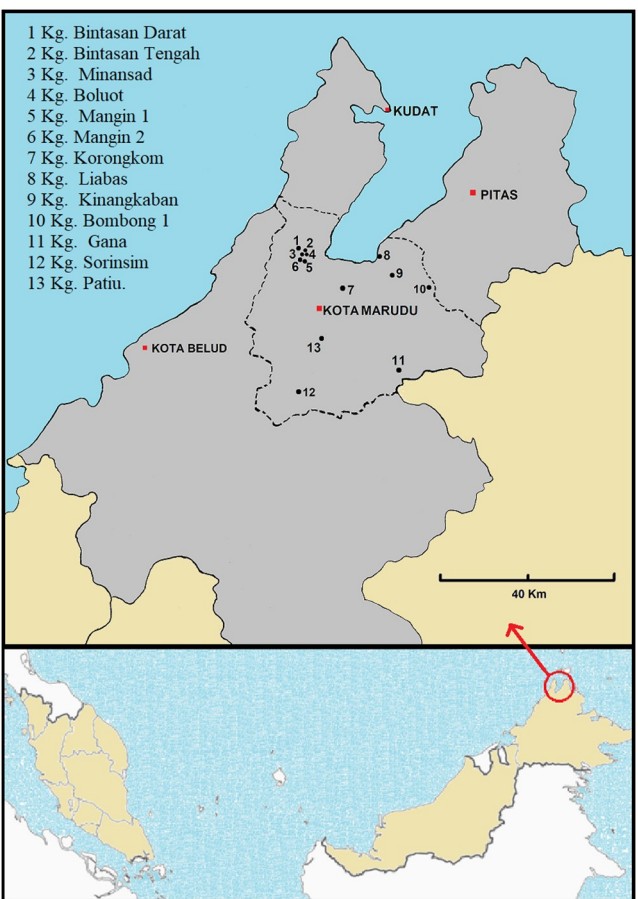

**Fig 1. Map of Kota Marudu district, Sabah where the study was conducted, indicating the location of the villages.**
The figure was modified using a map available from http://www.supercoloring.com/coloring-pages/malaysia-map, and
the village location plotted with Google earth software.

accessible by road, and had >20 houses with a population of >100 villagers. Ninety three vil-
lages fulfilled the criteria. The distance of the village from the Kota Marudu town and the ele-
vation were further considered in the selection to examine if the geohelminthiasis prevalence
was associated with these factors (Table 1). Permission to carry out study was sought from the
heads of the final list of 30 villages who were briefed on the purpose of the study. Due to time
and fund constraints, the first 13 villages with given permission were chosen as the combined
population provided a large sampling pool.

Although these villages are accessible by road, they are rather traditional and have very
basic social facilities and infrastructure. Despite the current development taking place in Kota
Marudu district, some of the villagers were found during the study, to live in poor environ-
mental sanitation conditions with no proper toilet facilities which resulted in some villagers
defecating in the bushes. Many households still rely on untreated rainwater for bathing and
washing, while some use gravity-feed or dug-well water. Only two villages have piped treated
water.

The population of the district in 2017 was 77,100, of which 27,240 were 17 years old and
below, consisting mainly Kadazan-Dusun (62%), Bajau (15%) and other native sub-ethnic
groups (17%) (Dept of Statistics, 2017). Within each village, all the children of age 6 months–
17 years old who were residing there were invited to participate in the study.

**Table 1. Details of the villages in Kota Marudu selected for the study.** Elevation of the villages is given as metres above sea level (masl). Total participants in the study were 407.

| Villages | Population | Distance from Kota Marudu Town (km) | GPS Coordinates | | Eleva-tion (m asl) | Total participants |
|---|---|---|---|---|---|---|
| Kg. Bintasan Darat | 214 | 23 | 6.5878 N | 116.6909 E | 43 | 46 |
| Kg. Bintasan Tengah | 288 | 20 | 6.5844 N | 116.7031 E | 43 | 30 |
| Kg. Minansad | 265 | 25 | 6.5692 N | 116.7021 E | 48 | 13 |
| Kg. Boluot | 115 | 25 | 6.5704 N | 116.704 E | 31 | 17 |
| Kg. Mangin 1 | 299 | 14 | 6.5543 N | 116.6912 E | 55 | 35 |
| Kg. Mangin 2 | 318 | 17 | 6.5589 N | 116.6817 E | 118 | 23 |
| Kg. Korongkom | 650 | 5 | 6.50765 N | 116.7767 E | 13 | 25 |
| Kg. Liabas | 618 | 20 | 6.56921 N | 116.8698 E | 168 | 35 |
| Kg. Bombong 1 | 443 | 30 | 6.5163 N | 116.9716 E | 49 | 11 |
| Kg. Gana | 361 | 38 | 6.3391 N | 116.9044 E | 715 | 71 |
| Kg. Kinangkaban | 410 | 28 | 6.5441 N | 116.9025 E | 279 | 63 |
| Kg. Patiu | 231 | 10 | 6.4419 N | 116.7374 E | 260 | 23 |
| Kg. Sorinsim | 158 | 30 | 6.2943 N | 116.7111 E | 180 | 15 |

## Sampling size

The required sampling size for the study was estimated by the formula:

$$n = z^2 p(1-p)/d^2 = 384;$$

where z = 1.96; p = proportion of the population infected = 0.5 [10]; d = level of significance = 0.05. Assuming a non-response rate of 5%, the targeted sampling size was 384/ 0.05 = 404.

## Collection of demographic data

Household data was collected through interviewing the children's parents or guardians in the respondent's house using structured questionnaires (S1 Questionnaire). The questionnaire was administered verbally in the Malaysian language and the answers recorded on paper.

The questions included parents' education background, occupation, source of drinking water, standard of hygiene practice (eg whether washing hands before meals, or after going to the toilet), consumption of raw meat, or aquatic vegetables, keeping pets in the house, allowing domestic animals (including pigs and goats) into the house, whether the children go barefoot, geophagy among the children, household waste disposal method etc. The quality of water supply, sanitation facilities, type and structure of house, and general cleanliness of the house environment were also recorded.

## Collection of faecal samples

All the households in each selected village were given a participant information sheet and then briefed on the objectives of the study. In addition, each household was provided with pre-labelled, clean, dry, leak-proof plastic, screw-capped stool containers, one bottle per child. The parents and older children were briefed on the procedure of stool collection, to ensure they adhere to the correct procedure for collecting the faecal samples. The parents/guardians of each respondent also signed an informed consent form in the Malaysian language. Confidentiality of participants was ensured throughout the study. To ensure a sufficient final sample size, a total of 500 bottles were given out.

## Examination of faecal samples

A direct wet mount of each faecal sample was made and examined by microscopy for the presence of geohelminth eggs in the laboratory of the district hospital within the same day. Egg count for positive samples was however not performed. After microscopy, all samples were then fixed with 10% formalin for further analysis at the laboratory of Pathobiology and Diagnostic Department, Universiti Malaysia Sabah. In the university laboratory, the negative samples were reexamined using formalin-ether concentration method (FEC) [15].

The geohelminths were initially identified based on the morphology of the eggs. Subsequent confirmation of *A. lumbricoides* and speciation of hookworm was done using polymercase chain reaction with the following primers. For *A. lumbricoides* the primer pair was EU582499 Fw: 5'-GGAGGTTTTTGGGTCTTTGG-3' and Rw: 5'-CCAAACAAGGTAGCCAACCA-3' [16]. For *Necator americanus*, the primers were AJ001680 Fw: 5'ATGCTTGGCAAGAGTCG TTT 3' and Rw: 5'TGTTGGCGTCCACACATATT 3' [16] and for *Ancylostoma* species, the primers were Ad125F: 5'GAATGACAGCAAACTCGTTGTTG 3' and Ad195R: 5'ATACTAG CCACTGCCGAAACGT 3' [17].

## Collection of soil and water samples

If the microscopy examination of the faecal samples indicated worm infection in the children, soil samples were taken for examination within four weeks after checking the faecal samples. Soil samples were taken from 40 households which had sandy or loamy house compounds. Within the compound where the children frequently played, two samples each about 200 g of soil were randomly collected with an ethanol-treated hand shovel at 5 cm beneath the surface of the soil which was not exposed to direct sunlight, placed in a new Quart Zipper Storage polyethylene bag and labelled accordingly.

The soil samples were processed in the laboratory using modified flotation method [18] with zinc sulphate (ZnSO4) solution of specific gravity 1.18–1.20. About 100 mL of distilled water was first added to 10 g of soil which was then homogenized with a glass rod before straining it through gauze and net mesh to remove large particles/stones. The strained mixture was then poured into two 50 mL conical tubes and centrifuged at 2500 rpm for one minute (Kubota 4000 Laboratory Centrifuge). The supernatant fluid was discarded, saline was added to the sediment and centrifuged. This process was repeated 2–3 times until the supernatant was clear. Approximately 5 mL of zinc sulphate (ZnSO4) was then added into the sediment and the mixture vortexed at 2200 rpm to loosen the pellet/sediment before being transferred into a 15 mL centrifuge tube. More ZnSO4 solution was added to the tube to make up to 15 mL, which was centrifuged again for five minutes at 1500 rpm (Hettich, Universal 320 laboratory 75 centrifuge). A coverslip was later placed on the top of the tube for 10–15 minutes to provide enough time for the STH eggs / larvae (if any) to float up and attach to the coverslip. The cover slip with eggs stuck to it was removed and examined under the microscope.

In each village which had children with geohelminth infection, two water samples were also collected from the river at the areas where the children reportedly always played or bathed or washed clothes. In the laboratory, the water sample was transferred to 15 mL screw-lid centrifuge tubes and centrifuged at 1500–2000 rpm for 5–10 minutes. The supernatant fluid was carefully decanted till 0.5 mL was left. The sediment was resuspended in the remaining supernatant before emptying it onto a glass slide and examined under the microscope for presence of geohelminth eggs.

## Statistical analyses

The relationship between various variables related to socio-economic factors and geohelminthiasis was examined with Pearson's $\chi^2$ test (SPSS version 22, Chicago, IL, USA) to identify the potential risk factors. This was followed by logistic regression to determine the odds ratios (OR) and 95% confidence interval (CI) (z = 1.96).

The geohelminthiasis infection data was further analyzed for clustering effect with the R package "Cluster Bootstrap" (https://github.com/mathijsdeen/ClusterBootstrap) [19] using R version 3.5.2 [20]. This package fits a generalized linear model (GLM) with cluster bootstrapping for analysis of clustered data. This was done separately for the number of children infected with (a) geohelminthiasis, and (b) ascariasis as a function of the various measured variables. Each village was considered as a cluster, while income, mother's education, availability of proper sanitation facility, household income, general hygiene etc, were treated as fixed factors. In each run, 5,000 bootstrap replicates were used. Clustering effect of infection within families was also similarly investigated by considering each family as a cluster and running GLM with 5,000 cluster bootstrap replicates.

## Results

### Demographic data

Fresh stool samples were collected from a total of 407 children (201 f and 206 m) aged between 6 months -17 years of age from 238 households. The response rate was 81% (407/500) and non-response reasons included constipation, having gone to the toilet the previous night or having lost the collection bottle.

There were 15 participants who were less than one year old, while the 4–6 and 7–12 year-olds formed the largest groups (26.5 and 41.3% respectively) (Table 2). The sex-ratio was about equal. About 83.8% of the respondent's mothers were housewives with no education (27.8%), primary (32.7%) or secondary education (36.6%), while only 8 had attended college/university. More than half (57%, 232/407) had monthly household income of less than US$119 (RM500, $1 = RM4.20) and 65.1% (265/407) stayed in one-storey stilted wooden house. Only 15% (45/407) had treated water where as the rest used untreated water from sources such as gravity-feed, dug well, river or rainwater. Most of the respondents (74.9%) still used pour flush toilets. Overall only about half of the children (49.9%) were reported to habitually wash their hands with soap. More children claimed they did not go barefoot (59.4%), did not wash their feet (77.2%), and did not trim their nails regularly (77.6%). However, more children had taken deworming medicine at least once (62.7%).

### Prevalence of geohelminthiasis

The FEC method gave a higher estimate of geohelminthiasis than direct wet faecal mount (14.3% compared to 10.6%) (Table 3). Overall, 9.6% of the sampled children had ascariasis, while 2.7% had either hookworm or *Trichuris* infection. *Ascaris lumbricoides* was the most common geohelminth detected in the infected children at 67.2% (or 62.1% for single-species infection), followed by hookworm (18.9%) and *Trichuris trichiura* (18.9%).

The mean infection rate among the children per village was 9.0% (0%-34.9%) or 16.8% (n = 7) considering only the villages with infected children. High prevalence of geohelminthiasis was detected in three villages, namely Kg. Kinangkaban (34.9%), Kg. Gana (25.4%) and Kg. Bintasan Darat (17.4%) (Table 4). However, geohelminthiasis was not detected in six villages. A total of 54% (7/13) of villages had children infected with worm infection. Overall, the mean

**Table 2. Socio-demographic data of participants in this study.** Total respondents = 407.

| Parameter | classes | N (%) |
|---|---|---|
| Age | 6–11 months | 15 (3.7) |
| | 1–3 years old | 83 (20.4) |
| | 4–6 years old | 108 (26.5) |
| | 7–12 years old | 168 (41.3) |
| | 13–17 years old | 33 (8.1) |
| Gender | Female | 201 (49.4) |
| | Male | 206 (50.6) |
| Mother's Level of Education | None | 113 (27.8) |
| | Pre-school | 4 (1.0) |
| | Primary | 133 (32.7) |
| | Secondary | 149 (36.6) |
| | College/University | 8 (1.9) |
| Monthly Household Income | Less than USD 119 | 232 (57.0) |
| | More than USD 119 | 175 (43) |
| People per household | Less than 5 people | 82 (20.1) |
| | More than 5 people | 325 (79.9) |
| Type of House | One-storey house with stilt | 265 (65.1) |
| | One-storey house on the ground | 104 (25.6) |
| | Long house | 3 (0.7) |
| | Double-storey house | 35 (8.6) |
| Source of water | Untreated water | 362 (88.9) |
| | Treated water | 45 (11.1) |
| Types of Toilet | No toilet | 39 (9.6) |
| | Flush toilets | 41 (10.1) |
| | Pour flush toilets | 305 (74.9) |
| | Pit Latrine | 22 (5.4) |
| Wash hands with Soap before eating | No | 204 (50.1) |
| | Yes | 203 (49.9) |
| Barefooted | Yes | 158 (40.6) |
| | No | 231 (59.4) |
| Washing of Feet | No | 312 (77.2) |
| | Yes | 92 (22.8) |
| Trimming of Nails | Yes | 90 (22.4) |
| | No | 311 (77.6) |
| Deworming | No or not sure | 152 (37.3) |
| | Yes | 255 (62.7) |
| With domestic animals | Dogs | 262 (64.6%) |
| | Cats | 222 (54.5%) |
| | Poultry | 187 (45.9%) |
| | Pigs | 16 (3.9%) |
| | Cattles | 14 (3.4%) |
| | Goats | 12 (2.9%) |

rate of infection of the children per village was 38.5% for *Ascaris*, 38.5% for hookworms and 30.8% for *Trichuris*. In Kg. Kinangkaban, 21/22 infections were ascariasis. Out of 238 households sampled, 43 (18%) had children with geohelminthiasis. The mean percentage of households with infected children in a village was 10.9±8.6%. (Table 5).

**Table 3. Prevalence of geohelminthiasis among the children in Kota Marudu.** Presence of geohelminths was detected by direct wet mount and formalin-ether concentration methods. Sample size = 407.

| | Method | | % of sampled children |
|---|---|---|---|
| | **Direct Wet Mount** | **Formalin-Ether Concentration** | |
| Negative samples | 364 (89.4%) | 349 (85.7%) | |
| Positive samples | 43 (10.6%) | 58 (14.3%) | |
| | Among the positive samples | | |
| *Ascaris lumbricoides* | 31(72.1%) | 36(62.1%) | |
| Hookworm | 7(16.3%) | 10(17.2%) | |
| *Trichuris trichiura* | 4(9.3%) | 9(15.5%) | |
| *Ascaris* + hookworm | 1(2.3%) | 1(1.7%) | |
| *Ascaris* + *Trichuris* | 0 | 2(3.4%) | |
| *Total infected with** | | | |
| *Ascaris* | 32 (74.4%) | 39 (67.2%) | 9.6% |
| *hookworm* | 8 (18.6%) | 11(18.9%) | 2.7% |
| *Trichuris* | 4 (9.3%) | 11(18.9%) | 2.7% |

* including multispecies infection

Of the 126 households which had ≥2 children per household, infection of ≥2 children in the same household was observed only in Kg Kanangaban which had six households (6/12) with more than one child infected with *Ascaris* (Table 5).

## Association analysis for risk factors

Using univariate analysis, the following factors: age (odds ratio, OR = 0.586) gender (OR = 0.745), the number of people in the household (OR = 1.812), type of house (OR = 1.316), burning of household waste (OR = 1.115), not trimming nails (OR = 1.281),

**Table 4. Prevalence of geohelminthiasis in each village in Kota Marudu.**

| Villages | Popula-tion | No. samples | No. Samples with eggs of | | | total (%) |
|---|---|---|---|---|---|---|
| | | | *Ascaris* | Hookworm | *Trichuris* | |
| Kg. Bintasan Darat | 214 | 46 | 4 | 3 | 1 | 8 (17.4%) |
| Kg. Bintasan Tengah | 288 | 30 | - | - | - | 0 |
| Kg. Minansad | 265 | 13 | - | - | - | 0 |
| Kg. Boluot | 115 | 17 | - | 3 | - | 3 (17.6%) |
| Kg. Mangin 1 | 299 | 35 | 1 | 3 | - | 3* (8.6%) |
| Kg. Mangin 2 | 318 | 23 | - | - | - | 0 |
| Kg. Korongkom | 650 | 25 | 2 | - | - | 2 (8.0%) |
| Kg. Liabas | 618 | 35 | - | - | 2 | 2 (5.7%) |
| Kg. Bombong 1 | 443 | 11 | - | - | - | 0 |
| Kg. Gana | 361 | 71 | 11 | 1 | 7 | 18* (25.4%) |
| Kg. Kinangkaban | 410 | 63 | 21 | 1 | 1 | 22* (34.9%) |
| Kg. Patiu | 231 | 23 | - | - | - | 0 |
| Kg. Sorinsim | 158 | 15 | - | - | - | 0 |
| % village with geohelminth detected | | | 38.5% | 38.5% | 30.8% | Mean/village9.0% (n = 13) |
| | | | | | | 16.8% (n = 7) |

* denotes multiple infection by more than one species of geohelminth

**Table 5. Geohelminthiasis prevalence among the children within each household.**

| Village | total households sampled | Households with 1 child | | Households with ≥2 children | | | | Total households with Infected children | % of households with infected children |
|---|---|---|---|---|---|---|---|---|---|
| | | number | Infected | number | 1 child infected | ≥2 infected | Total infected | | |
| Kg Bombong 1 | 10 | 9 | 0 | 1 | 0 | 0 | 0 | 0 | 0.0 |
| Kg Bintasan Darat | 25 | 8 | 1 | 17 | 4 | 1 | 5 | 6 | 24.0 |
| Kg Boluot | 7 | 1 | 0 | 6 | 1 | 0 | 1 | 1 | 14.3 |
| Bintasan Tengah | 16 | 7 | 0 | 9 | 0 | 0 | 0 | 0 | 0.0 |
| Kg Gana | 37 | 13 | 3 | 24 | 11 | 2 | 3 | 16 | 43.2 |
| Kg Korongkom | 14 | 6 | 0 | 8 | 0 | 1 | 1 | 1 | 7.1 |
| Kg Kinangkaban | 41 | 21 | 4* | 20 | 6* | 6* | 12 | 16 | 39.0 |
| Kg Liabas | 19 | 11 | 0 | 8 | 1 | 0 | 1 | 1 | 5.3 |
| Kg Mangin 1 | 23 | 14 | 0 | 9 | 1 | 1 | 2 | 2 | 8.7 |
| Kg Mangin 2 | 16 | 10 | 0 | 6 | 0 | 0 | 0 | 0 | 0.0 |
| Kg Minansad | 6 | 1 | 0 | 5 | 0 | 0 | 0 | 0 | 0.0 |
| Kg Sorinsim | 9 | 4 | 0 | 5 | 0 | 0 | 0 | 0 | 0.0 |
| Kg Patiu | 15 | 7 | 0 | 8 | 0 | 0 | 0 | 0 | 0.0 |
| total | 238 | 112 | 8 | 126 | 14 | 11 | 25 | 43 | 10.9±8.6 |

\* all children were infected with *Ascaris*

keeping domestic animals in the house, and deworming (OR = 1.121) were not risk factors of geohelminthiasis (Table 6).

The following factors were found significantly associated with geohelminth infections: mother's lack of education (22.1% vs 11.2% infection for mothers with at least kindergarten education, p = 0.003) (Table 6), monthly household income of <USD119 (<RM500) (21.5% vs 4.6% infection for >USD119, <0.001), using untreated water (15.5% vs 4.4%, p = 0.046), lack of proper sanitation facilities (46.2% vs 10.9%, p = <0.001), not washing hands with soap before meals (17.6% vs 10.8%, p = 0.049), going barefoot outside the house (22.2% vs 9.5%, p = 0.001), and not washing feet before entering the house (16.3% vs 6.5%, p = 0.017). Additionally, children walking barefooted was a significant factor of hookworm infection ($X^2$ = 4.84, df = 1, p = 0.028).

Similarly logistic regression analysis indicated the following risk factors of geohelminthiasis: mothers lacking education (OR = 0.445, p = 0.006); household income of less than USD119 (OR = 0.174, p<0.0001); lack of proper sanitation facilities (OR = 0.142, p<0.0001); children who go barefoot (OR = 0.370, p = 0.001) and children not washing the feet (OR = 0.357, p = 0.022) (Table 7).

## Analysis of clustering of infection

Analysis using generalized linear model with cluster bootstrapping was done to see if there was clustering effect at the village level, since only Kg Gana and Kg Kinangkaban had high infection rate. However, bootstrapping was not possible with two variables, viz. treated water source (ie piped water) versus untreated (rainfall, well, river) for both drinking and for other uses. This is because only two children from the two villages with treated water supply (all households in Kg Korongkom and 6/14 in Kg Bintasan Tengah) had geohelminth infection. This

**Table 6. Association analysis of factors with geohelminthiasis (univariate analysis).** Data based on 407 children in 13 rural villages in Kota Marudu). Missing data were omitted from the analysis.

| Factors | N (%) | Positive samples | Prevalence (%) | $X^2$ value (p-value) | Odds ratio (95% CI) |
|---|---|---|---|---|---|
| Age group | | | | | |
| 12 years and below | 374 (91.9%) | 51 | 13.6 | 1.424 | 0.586 |
| 13 years and above | 33 (8.1%) | 7 | 21.2 | (0.233) | (0.242–1.421) |
| Gender | | | | | |
| Male | 206 (50.6%) | 33 | 16.0 | 1.068 | 0.745 |
| Female | 201 (49.4%) | 25 | 12.4 | (0.301) | (0.425–1.304) |
| Mother's Education | | | | | |
| None | 113 (27.8%) | 45 | 22.1 | 8.546 | 2.317 |
| Yes | 294 (72.2%) | 13 | 11.2 | (0.003)* | (1.315–4.113) |
| Monthly household income | | | | | |
| < RM500 (<USD119) | 232 (57.0%) | 50 | 21.6 | 23.537 | 5.735 |
| > RM500 | 175 (43.0%) | 8 | 4.6 | (<0.001)* | (2.641–12.452) |
| People per household | | | | | |
| < 5 people | 82 (20.1%) | 17 | 20.7 | 3.530 | 1.812 |
| > 5 people | 325 (79.9) | 41 | 12.6 | (0.06) | (0.968–3.389) |
| Type of House | | | | | |
| One-storey house with stilt | 372 (91.4%) | 54 | 14.5 | 0.250 | 1.316 |
| Others | 35 (8.6%) | 4 | 11.4 | (0.617) | (0.447–3.877) |
| Source of Water | | | | | |
| Untreated water | 362 (88.9%) | 56 | 15.5 | 3.981 | 3.935 |
| Treated water | 45 (11.1%) | 2 | 4.4 | (0.046)* | (0.927–16.709) |
| Proper sanitation facility | | | | | |
| None | 39 (9.6%) | 18 | 46.2 | 35.927 | 7.029 |
| Yes | 368 (82%) | 40 | 10.8 | (<0.001)* | (3.456–14.296) |
| Household Waste | | | | | |
| Burning | 309 (75.9%) | 45 | 14.6 | 0.103 | 1.115 |
| Not burning | 98 (24.1%) | 13 | 13.3 | (0.749) | (0.574–2.165) |
| Wash hands with Soap before meals | | | | | |
| No | 204 (50.1%) | 36 | 17.6 | 3.861 | 1.763 |
| Yes | 203 (49.9%) | 22 | 10.8 | (0.049)* | (0.997–3.119) |
| Barefooted | | | | | |
| Yes | 158 (40.6%) | 35 | 22.2 | 11.964 | 2.703 |
| No | 231 (59.4%) | 22 | 9.5 | (0.001)* | (1.517–4.818) |
| Washing of Feet | | | | | |
| No | 312 (77.2%) | 51 | 16.3 | 5.659 | 2.801 |
| Yes | 92 (22.8%) | 6 | 6.5 | (0.017)* | (1.161–6.754) |
| Trimming of Nails | | | | | |
| Yes | 90 (22.4%) | 15 | 16.7 | 0.572 | 1.281 |
| No | 311 (77.6%) | 42 | 13.5 | (0.449) | (0.674–2.436) |
| Deworming status | | | | | |
| No or Not sure | 152 (37.3%) | 23 | 15.1 | 0.154 | 1.121 |
| Yes | 255 (62.7%) | 35 | 13.7 | (0.695) | (0.634–1.980) |

* p-value < 0.05

**Table 7. Logistic regression analysis of risk factors associated with geohelminth infections among the children of 13 villages in Kota Marudu.**

| Factor | Odds ratio (OR) | 95% CI | P-value |
|---|---|---|---|
| Mother's lack of education | 0.445 | 0.251–0.789 | 0.006** |
| Household income of less than RM500 | 0.174 | 0.080–0.379 | 0.000*** |
| Using of untreated water as water source | 0.254 | 0.060–1.079 | 0.0063 |
| Do no wash hands with soap before meals | 0.567 | 0.321–1.003 | 0.051 |
| Availability of sanitation facilities | 0.142 | 0.070–0.289 | 0.000*** |
| Children who were barefooted | 0.370 | 0.208–0.659 | 0.001** |
| Not washing the feet | 0.357 | 0.148–0.861 | 0.022* |

* p-value < 0.05;

**p-value < 0.01;

*** p-value < 0.001

results in a lack of variation in the outcome variable and no contrast was possible in the GLM run. Similarly, bootstrapping with sanitation facility was not possible as only 2/4 villages (Kg Bintasan Darat and Kg Kinangkaban) without sanitation facility had recorded geohelminth infection among the children (S1 Table).

Adjusting for the effect of infection clusters in the villages, only household income (≥USD119) had almost a significant impact on reducing total geohelminth infection (three species combined) (Table 8). The other factors (eg mother's education level, deworming, proper sanitation facility) had some positive but non-significant effect only. Adjusting for family clustering, GLM analysis bootstrapping indicated proper sanitation facility has a significant positive effect on reducing *Ascaris* infection ($P < 0.05$), but not for total worm infection (Table 9).

## Analysis of soil and water samples

Almost all the soil samples collected (36/40) from the house compounds had loamy texture (definition of Zenner et al. [21]). Geohelminths were detected in the soil samples of 14 houses

**Table 8. Analysis using generalized linear model with villages considered as clusters of geohelminthiasis.** Bootstrapping with 5000 replicates was performed. The parameter "proper sanitation" had been omitted since it has only one level in some villages.

| Parameter | Estimate | SE | z |
|---|---|---|---|
| Infection with *Ascaris* | | | |
| Intercept | -1.5428 | 1.1578 | |
| Mother's education level (none vs at least kindergarten) | -0.4160 | 0.3043 | -1.3672 |
| Household income (<RM 500/USD119 vs more) | -3.2462 | 4.4375 | -0.7315 |
| Use of soap in washing hands | -0.0841 | 0.6547 | -0.1285 |
| Washing feet before entering house (not washing vs washing) | -1.6849 | 4.0613 | -0.4149 |
| Deworming vs none | 0.1062 | 0.8632 | 0.1230 |
| Infection with all species | | | |
| Intercept | -0.9129 | 0.6613 | |
| Mother's education level (none vs at least kindergarten) | -0.1964 | 0.4390 | -0.4473 |
| Household income (<RM 500/USD119 vs more) | -1.7810 | 0.9105 | -1.9561 |
| Use of soap in washing hands | -0.5044 | 0.3467 | -1.4552 |
| Washing feet before entering house (not washing vs washing) | -1.2990 | 2.4698 | -0.5260 |
| Keeping of domestic animals vs none | 0.0278 | 0.5678 | 0.0489 |
| Deworming vs none | -0.1964 | 0.4390 | -0.4473 |

**Table 9. Analysis of factors of ascariasis using generalized linear model with families considered as clusters of geohelminthiasis.** Bootstrapping with 5000 replicates was performed.

| Parameter | Estimate | SE | z |
|---|---|---|---|
| Infection with *Ascaris* | | | |
| Intercept | 2.6369 | 1.6622 | |
| Mother's education level (none vs at least kindergarten) | -1.0310 | 1.1693 | -0.8817 |
| Household income (<RM 500/USD119 vs more) | -7.5320 | 7.8494 | -0.9595 |
| Use of soap in washing hands | -1.2254 | 1.2488 | -0.9813 |
| Washing feet before entering house (not washing vs washing) | -0.6683 | 2.7825 | -0.2402 |
| Deworming vs none | -0.7647 | 0.6812 | -1.1226 |
| Proper sanitation facility vs none | -3.5419 | 1.7360 | -2.0403* |

* $p < 0.05$

(35%), 12 of which had hookworm eggs, one sample had *Ascaris* eggs, and another had both hookworm and *Trichuris* eggs (Table 10). *Ascaris* eggs were found in the soil sample of a household in Kg. Kinangkaban whose child was also infected with *A. lumbricoides* (KK09). Similarly, *Trichuris* eggs were found in the soil sample taken from the house (GN47) which had the child infected with *Trichuris* in Kg. Gana. All the soil samples positive with STH eggs were of loamy texture.

In the analysis of water samples taken from the villages where worm-infected children were recorded, none of the samples was positive with any of the helminth eggs although unidentified larvae and algae were observed under the microscope.

## Analysis of geohelminth infection with altitude and distance of village from the main town

The percentage geohelminth infection increased, but not significantly, with both the distance from the main town (Kota Marudu), and the altitude of the village (see S1 Fig).

## Discussion

This is the first study on the prevalence rate of geohelminthiasis available in Kota Marudu, a fast-developing town. Our results show that geohelminth infections are still prevalent at 54% of the villages studied, 18% of the households sampled and 14.3% of the children examined.

**Table 10. Detection of geohelminth stages from soil samples.** Samples were taken from the compound of 40 households whose children's faecal samples had tested positive for worms.

| Village | Houses | *Ascaris* eggs | Hookworm eggs | *Trichuris* eggs |
|---|---|---|---|---|
| Kg. Bintasan Darat | 5 | 0 | 2 | 0 |
| Kg. Boluot | 1 | 0 | 0 | 0 |
| Kg. Gana | 14 | 0 | 7* | 1 |
| Kg. Korongkom | 1 | 0 | 0 | 0 |
| Kg. Kinangkaban | 15 | 1 | 2 | 0 |
| Kg. Liabas | 2 | 0 | 2 | 0 |
| Kg. Mangin 1 | 2 | 0 | 0 | 0 |
| Total | 40 | 1 | 13 | 1 |

* indicates a soil sample from one household (GN47) had both hookworm and *Trichuris* eggs

Proportion of ascariasis among the infected children was 67.2%, hookworm 18.9% and *Trichuris trichiura* 18.9%.

The risk factors are maternal lack of education, low household income (<USD119), lack of proper sanitation facility, not wearing footwear, washing feet, and washing hands before meals. The factors which were found not affecting geohelminthiasis are age, gender, the number of people in the household, type of house, burning of household waste, untrimmed nails and deworming.

The geohelminthiasis prevalence recorded (14.3%) was almost similar to a previous study in 2003 among the communities in the fringes of the Crocker Range Park, Sabah at 14.7% [14] which is about 144 km away by road. This seems to indicate that the prevalence has not changed much over the years. Our research recorded prevalence rates among the children for *Ascaris*, *Trichuris* and hookworm of 9.6%, 2.7% and 2.7% compared to 8.7%, 10% and 3.3% for all age groups (up to >31 years old) [14]. In West Malaysia, the prevalence reported for aboriginal children of Pos Sungai Rual, Kelantan was much higher at 40.5%, 65.8% and 25.8% respectively [22]. The Sabah prevalence rates are comparable to those recorded in neighbouring Sarawak in two studies on children with the respective rates of 7–12.8%, 28.4% and 7.2% [23, 24] The prevalence of *Trichuris* in Kota Marudu villages appears to be lower compared to the other Malaysian study sites, but we are unable to ascertain why.

Taking clustering effect at the village level, GLM analysis indicated low income of ≤RM500 (USD119) is an important factor of *Ascaris* infection, whereas at the family level, lack of sanitation facilities is a significant risk factor.

Some maternal formal education was shown to be a positive factor. This is in concordance with the findings by other workers. In Sri Lanka, poor maternal education was identified as a risk factor for infection [25], and in Bihar, India the mother's literacy was found significantly associated with geohelminth infections [26]. Low literacy has somehow affected the level of hygiene practiced in a household and has also resulted in low household income and related health problems. Furthermore, we showed that, considering the clustering effect at the village level, receiving some education does have some positive effect. Among the households where mothers had no formal education, washing hands with soap appears to be more important: 18/66 (38%) of children who did not wash their hands with soap were infected compared to 7/47 (18%) of those who used soap.

Higher income (>USD 119) has a positive effect on reducing geohelminthiasis, and has clustering effect at the village level. Low household income was found to be negatively associated with geohelminthiasis in another study among the indigenous children (*Orang Asli*) in Lipis, Pahang, West Malaysia [27]. Low income may result in reluctance to build proper sanitation facilities, and defecation in the open space, bushes, field or jungle is commonly practiced in poor villages, which could cause many environmental and health problems. This could partly explain the prevalence of STH in these villages where hygiene is poor, lacking sanitation facilities and treated water.

Geohelminthiasis is mostly transmitted when faeces containing eggs are deposited into the environment especially through open defecation, and ingested (eg. *Ascaris*) or transmitted across the skin boundary (eg hookworm) [28–30]. Our results indicate 35% of the soil samples taken from the houses with infected children had geohelminth eggs, especially hookworm. Higher percentages were recorded in an indigenous village in west Malaysia, namely 90% of 40 soil samples tested positive for *A. lumbricoides* and 15% were positive for *T. trichiura* [31]. Contaminated soil in these villages is a risk factor in geohelminthiasis, as evident from our results that the children going barefoot outside the house and not washing feet before entering the house were both significantly associated with geohelminth infections, especially hookworm infection (p = 0.028). Similarly, villagers in Thailand, including children walking barefoot was

also a significant factor of hookworm infection [32]. Regular deworming with benzimidazole anthelmintic drugs in school-age children has been shown to reduce and maintain the worm burden below the threshold associated with disease [33]. However, we are not able to establish deworming as a positive risk factor in our study (OR = 1.121, p>0.05), probably because the deworming was inconsistent and not all the children were taking deworming medicine. Although the Ministry of Health offers free deworming services, only 68% of the villagers confirmed their children had taken the deworming medicine at least once, sometime in the past, while some parents could not recall if their child had taken any. The questionnaire results also revealed that only l219/407 (54%) of the children took deworming medicine on the advice of the doctor. Nevertheless, the importance of deworming cannot be discounted.

The geohelminthiasis prevalence is a public health concern among the indigenous children of Kota Marudu. Studies have shown that in children with geohelminth infections psychological and physical development were delayed, and adversely affecting their ability to participate in life at school and at home. Such delays limit their ability to take full advantage of what is often their only opportunity for formal education and may limit their social functioning later in life [4].

In conclusion, this study clearly shows that geohelminthiasis still occurs with an overall prevalence of 14.3% among children of rural communities of Kota Marudu in Sabah. Infection by *Ascaris* had the highest rate at 67% compared to hookworm or *T. trichiura*. The most practical means of controlling geohelminthiasis in these communities would be expanding mass deworming programmes for school-age children, improving sanitation and water supplies, upgrading the socio-economic status and continually conducting health education programmes. The data on the prevalence of geohelminthiasis in this study would contribute to better public health monitoring and operation to reduce the infection in rural areas.

## Supporting information

**S1 Questinnaire. Sample pages of the questionnaire used in the study.**
(PDF)

**S1 Table. Distribution of geohelminth-infected children in the villages among the households with or without basic sanitation facilities.**
(PDF)

**S1 Fig. Regression of percentage geohelminthiasis among the children against (a) distance (km) of village from the main town (Kota Marudu), and (b) altitude of village (metres above sea level).**
(TIF)

## Acknowledgments

The authors wish to acknowledge the cooperation and support from the village heads and communities of Kota Marudu for assisting in conducting this research successfully. The authors would also like to thank Universiti Malaysia Sabah for the research facilities, and the Director, Science Officer, Senior Medical Laboratory Technologists and Medical Laboratory Technologists and staff of Kota Marudu Hospital for granting access to their laboratory and facilities.

## Author Contributions

**Conceptualization:** A. Lim-Leroy, Tock H. Chua.

**Data curation:** A. Lim-Leroy.

**Formal analysis:** Tock H. Chua.

**Funding acquisition:** Tock H. Chua.

**Investigation:** A. Lim-Leroy.

**Methodology:** Tock H. Chua.

**Project administration:** Tock H. Chua.

**Resources:** Tock H. Chua.

**Supervision:** Tock H. Chua.

**Validation:** Tock H. Chua.

**Visualization:** Tock H. Chua.

**Writing – original draft:** A. Lim-Leroy.

**Writing – review & editing:** Tock H. Chua.

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
