## [Decision Letter · Decision Letter 0]

23 Jul 2020

PONE-D-20-17984

Prevalence and risk factors of geohelminthiasis among the rural village children in Kota Marudu, Sabah, Malaysia

PLOS ONE

Dear Dr. Chua,

Thank you for submitting your manuscript to PLOS ONE. After careful consideration, we feel that it has merit but does not fully meet PLOS ONE’s publication criteria as it currently stands. Therefore, we invite you to submit a revised version of the manuscript that addresses the points raised during the review process.

We look forward to receiving your revised manuscript.

Kind regards,

Arun K Yadav, Ph.D.

Academic Editor

PLOS ONE

Journal Requirements:

Additional Editor Comments (if provided):

Thank you for submission of your manuscript. I have seriously gone through the reviewed article and could see that the reviewers have given a number of useful suggestions on your article. Kindly consider the raised points, which I think are truly worthy of consideration and can remarkably improve your manuscript. In particular, please look into the following suggestions very carefully:

- Please see that the criterion for selection of study area and sample size is very clear and is also appropriately justified in the light of reviewers' comments.

- Also please ensure that the introduction and discussion section of draft is not much generalized, instead focused, with proper highlights of major innovative findings of study.

Reviewers' comments:

Reviewer's Responses to Questions

**Comments to the Author**

1. Is the manuscript technically sound, and do the data support the conclusions?

Reviewer #1: Yes

Reviewer #2: Yes

2. Has the statistical analysis been performed appropriately and rigorously? 

Reviewer #1: Yes

Reviewer #2: Yes

3. Have the authors made all data underlying the findings in their manuscript fully available?

Reviewer #1: Yes

Reviewer #2: Yes

4. Is the manuscript presented in an intelligible fashion and written in standard English?

Reviewer #1: Yes

Reviewer #2: Yes

5. Review Comments to the Author

Reviewer #1: General observations:

The authors present a cross section of data on the prevalence and risk factors for STH in children from 13 rural villages of a district in Sabah, Malaysia. This information will be useful for implementing strategies for the control and prevention of STH in this population.

Specific comments:

1. Abstract, Author Summary and text:

‘Mean infection rate’ should be changed to ‘mean prevalence of infection’ - this study has determined only point prevalence of STH infection; infection rate refers to new infections over a period of time.

Lines 27 – 30 in Abstract: include P values for the significant variables.

2. Introduction: provide references for the poor outcomes mentioned with hookworm disease and pregnancy (lines 68-70).

Line 71: Trichuris trichiura (or T. trichiura) – better to provide the entire scientific name when referring to the worms.

3. Materials and Methods: Line 99 refers to the 13 villages selected for the study out of 138 based on certain criteria. Were these 13 villages the only ones which fulfilled these criteria? Or were there others and only these 13 were selected? If so, how was the selection done?

Examination of faecal samples (lines 150-151) – what was the time duration between sample collection and examination of the direct wet mount done at the district hospital?

What was the reason for doing molecular testing for the positive samples? Shouldn’t the negative ones have been tested instead? That would have provided more accurate data on the prevalence of infection as PCR is more sensitive than the other two methods used. Furthermore, formalin preservation is known to fragment DNA with time and not usually used if PCR is being considered. So how can the authors ensure that the extracted DNA was of good quality? In addition, no information regarding the PCR test results have been mentioned in the text. My suggestion is to either entirely omit the mention of molecular testing as it doesn’t appear to add value to this manuscript in its current state or include the relevant data obtained through PCR testing and discuss their limitations.

Line 196 mentions collection of soil samples from 40 households which were selected randomly from among the positive children. Table 5 shows the total number of infected households as 25. Please state clearly the basis for collection of the soil samples.

4. Results: Lines 243-245 have discrepancies in either (or both) the percentages and/or the numbers mentioned which need to be corrected – Eg. 57%, 118/507 – according to Table 2 it should be 232/407; similarly, 265/507 should be 265/407; and 15% (161/507) should be 11.1% (45/407).

Line 248 – states that more children go barefooted (59.4%) but Table 2 states differently.

Line 316 – X2 value has to be included

Line 358 – Table 10 heading should read as Hookworm eggs (unless larvae were also found)

5. Discussion:

Under introduction it is stated that globally Malaysia records one of the highest prevalence rates for Trichuris. It is also apparent from other studies done in the area. Do the authors have any explanations as to why the prevalence rate seen in this study is lower? Especially since both Ascaris and Trichuris have similar transmission routes and are known to be co-extensive.

Line 407 mentions that 35% of households sampled had geohelminth eggs or other stages in the soil – please state the other stages that were found as they are not mentioned in Table 10.

6. Grammar and typing errors:

Abstract:

Lines 19 & 31: delete ‘the’; Line 29: should be USD 119

Author Summary:

Line 41: delete ‘old’; Line 42: delete the second ‘the’ (children); Line 47: change to - USD, facilities. Line 48: replace ‘had’ with ‘were’, delete ‘been’; Line 50: programs; Line 51: sanitation facilities

Introduction: Line 57: delete ‘the’; Line 76: replace ‘has’ with ‘have’; Line 78: replace ‘had’ with ‘have’; Line 79; backgrounds; Line 89: change to ‘between the ages of 6-months to 17 years’ and delete ‘old’

Materials and Methods: Line 93: ‘carried out’; Line 94: replace ‘of’ with ‘and’; Line 97: modified ‘using’ a map; Line 102: geohelminthiasis; Line 136: ‘washing’ hands, ‘the toilet’; Line 162: ‘microtube’; Line 179: Kuala ‘Lumpur’; Line 182: DNA ‘sequence’ results; Line 183: ‘databases’; Line 187: ‘amplify’; Line 222: ‘Pearson’s’

Results: Line 243: 8 had ‘attended’ college/university; Line 258: ‘Socio-demographic’; Line 265: remove ‘only’; Line 275: should read as ‘villages had children infected with’ worm infection, also delete ‘children per’ and change village to ‘villages’; Line 276: ‘infections were’; Line 315: 6.5%’,’ and delete ‘villagers including’ and change to ‘barefooted’; Line 317: logistic regression ‘analysis’; Line 319: ‘facilities’; Line 351: Ascaris ‘eggs’; Line 352: Trichuris ‘eggs’, Ascaris ‘eggs were’; Lines 353 -354: should read as ‘Similarly, Trichuris eggs were’; Line 359: (GN47) ‘which’ had, Trichuris ‘eggs’; Line 364: replace ‘was’ with ‘were’

Discussion: Line 372: ‘show’; Line 376: facility,; Line 382: should read as ‘Our research recorded prevalence rates’; Line 386: delete first ‘the’ (children); Line 391: ‘concordance’; Line 397: ‘hands’; Line 398: ‘wash their hands’; Line 400: ‘geohelminthiasis’; Line 403: sanitation ‘facilities’; Line 408: delete ‘were’; Line 409: 15% were positive ‘for’ T. trichura; Line 419: delete ‘periodically’; Line 431: delete ‘”’; Line 432: ‘practiced by the villagers’; Line 433: delete first ‘the’; Line 439: replace ‘of’ with ‘to their’; Lines 439-440: ‘trained at a very young age on the use of the toilet and wearing footwear outside the house’

Acknowledgements: Line 449: replace ‘which’ with ‘wish’; Line 450: for assisting in ‘conducting’ this research successfully. delete ‘conducted’.

Reviewer #2: The authors have discussed the Prevalence and risk factors of geohelminthiasis among the rural village children in Kota Marudu, Sabah, Malaysia. Of the 13 villages surveyed, geohelminthiasis was found prevalent only in 7, while there was no infection reported among the subjects of 6 villages. Though the authors have carried out extensive statistical analyses of data, the sample size in the study seems rather small. The results obtained through this study appear to be a corroboration of earlier studies carried out elsewhere across the globe.

[Eg., Prevalence and risk factors of soil-transmitted helminthiasis ...

bmcpublichealth.biomedcentral.com › articles;

Prevalence & risk factors for soil transmitted helminth infection ...

www.ncbi.nlm.nih.gov › pmc › articles › PMC3994744;

Prevalence and risk factors associated with the presence of ...

www.ncbi.nlm.nih.gov › pmc › articles › PMC4554591;

Prevalence & risk factors for soil transmitted helminth ...

www.researchgate.net › publication › 260610263_Prevalence_risk_fact.;

Prevalence and risk factors associated with worm infestation ...

www.researchgate.net › publication › 4181077_Prevalence_and_risk_f...;

Geohelminth Infections and Nutritional Status of Preschool ...

www.hindawi.com › journals › scientifica;

Prevalence, Intensity of Soil-Transmitted Helminths, and ...

www.hindawi.com › journals › jtm;

The prevalence, intensities and risk factors associated with ...

onlinelibrary.wiley.com › doi › full;

prevalence and risk factors associated with worm infestation in ...

www.bioline.org.br › pdf- to name a few].

I wish the authors had highlighted the novel findings emerging from their study and avoided a lengthy textbook account in ‘Introduction’ and generalizations in ‘Discussion’.

6. PLOS authors have the option to publish the peer review history of their article (what does this mean?). If published, this will include your full peer review and any attached files.

Reviewer #1: No

Reviewer #2: No

---

## [Author Response · Author response to Decision Letter 0]

21 Aug 2020

Reviewer #1: General observations:

The authors present a cross section of data on the prevalence and risk factors for STH in children from 13 rural villages of a district in Sabah, Malaysia. This information will be useful for implementing strategies for the control and prevention of STH in this population.

Specific comments:

1. Abstract, Author Summary and text:

‘Mean infection rate’ should be changed to ‘mean prevalence of infection’ - this study has determined only point prevalence of STH infection; infection rate refers to new infections over a period of time.

This has been done

Lines 27 – 30 in Abstract: include P values for the significant variables.

Done

2. Introduction: provide references for the poor outcomes mentioned with hookworm disease and pregnancy (lines 68-70).

Done

Line 71: Trichuris trichiura (or T. trichiura) – better to provide the entire scientific name when referring to the worms.

done

3. Materials and Methods: Line 99 refers to the 13 villages selected for the study out of 138 based on certain criteria. Were these 13 villages the only ones which fulfilled these criteria? Or were there others and only these 13 were selected? If so, how was the selection done?

How many fit criteria? 

This has been explained more clearly in the revised version (see lines 99-106 revised copy)

Examination of faecal samples (lines 150-151) – what was the time duration between sample collection and examination of the direct wet mount done at the district hospital?

Within same day.

What was the reason for doing molecular testing for the positive samples? Shouldn’t the negative ones have been tested instead? That would have provided more accurate data on the prevalence of infection as PCR is more sensitive than the other two methods used. Furthermore, formalin preservation is known to fragment DNA with time and not usually used if PCR is being considered. So how can the authors ensure that the extracted DNA was of good quality? In addition, no information regarding the PCR test results have been mentioned in the text. My suggestion is to either entirely omit the mention of molecular testing as it doesn’t appear to add value to this manuscript in its current state or include the relevant data obtained through PCR testing and discuss their limitations.

Thank you for pointing this out. You are right, the molecular results have not been fully shown here. I agree with you that it’s better to omit it. However I include the primers used in the id of eggs as information to interested readers.

Line 196 mentions collection of soil samples from 40 households which were selected randomly from among the positive children. Table 5 shows the total number of infected households as 25. Please state clearly the basis for collection of the soil samples.

Thank you for highlighting this. Table 5 has been relabelled and the error corrected. The number of infected children in households with >2 children should be “11” and not “1”. The total households with infected children was 43, while samples were taken from 40 houses. 3 houses were not suitable for sampling.

4. Results: Lines 243-245 have discrepancies in either (or both) the percentages and/or the numbers mentioned which need to be corrected – Eg. 57%, 118/507 – according to Table 2 it should be 232/407; similarly, 265/507 should be 265/407; and 15% (161/507) should be 11.1% (45/407).

Thank you for pointing out this. The errors have been corrected, along with other errors as well.

Line 248 – states that more children go barefooted (59.4%) but Table 2 states differently.

The error has been corrected.

Line 316 – X2 value has to be included

Done

Line 358 – Table 10 heading should read as Hookworm eggs (unless larvae were also found)

done

5. Discussion:

Under introduction it is stated that globally Malaysia records one of the highest prevalence rates for Trichuris. It is also apparent from other studies done in the area. Do the authors have any explanations as to why the prevalence rate seen in this study is lower? Especially since both Ascaris and Trichuris have similar transmission routes and are known to be co-extensive.

This is a very good point, but we don’t know the answer. A sentence has been added to state this.

Line 407 mentions that 35% of households sampled had geohelminth eggs or other stages in the soil – please state the other stages that were found as they are not mentioned in Table 10.

Only eggs were detected. This has been corrected

6. Grammar and typing errors:

All grammatical errors pointed out in the following sections have been corrected .

Abstract:

Lines 19 & 31: delete ‘the’; Line 29: should be USD 119

Author Summary:

Line 41: delete ‘old’; Line 42: delete the second ‘the’ (children); Line 47: change to - USD, facilities. Line 48: replace ‘had’ with ‘were’, delete ‘been’; Line 50: programs; Line 51: sanitation facilities

Introduction: Line 57: delete ‘the’; Line 76: replace ‘has’ with ‘have’; Line 78: replace ‘had’ with ‘have’; Line 79; backgrounds; Line 89: change to ‘between the ages of 6-months to 17 years’ and delete ‘old’

Materials and Methods: Line 93: ‘carried out’; Line 94: replace ‘of’ with ‘and’; Line 97: modified ‘using’ a map; Line 102: geohelminthiasis; Line 136: ‘washing’ hands, ‘the toilet’; Line 162: ‘microtube’; Line 179: Kuala ‘Lumpur’; Line 182: DNA ‘sequence’ results; Line 183: ‘databases’; Line 187: ‘amplify’; Line 222: ‘Pearson’s’

Results: Line 243: 8 had ‘attended’ college/university; Line 258: ‘Socio-demographic’; Line 265: remove ‘only’; Line 275: should read as ‘villages had children infected with’ worm infection, also delete ‘children per’ and change village to ‘villages’; Line 276: ‘infections were’; Line 315: 6.5%’,’ and delete ‘villagers including’ and change to ‘barefooted’; Line 317: logistic regression ‘analysis’; Line 319: ‘facilities’; Line 351: Ascaris ‘eggs’; Line 352: Trichuris ‘eggs’, Ascaris ‘eggs were’; Lines 353 -354: should read as ‘Similarly, Trichuris eggs were’; Line 359: (GN47) ‘which’ had, Trichuris ‘eggs’; Line 364: replace ‘was’ with ‘were’

Discussion: Line 372: ‘show’; Line 376: facility,; Line 382: should read as ‘Our research recorded prevalence rates’; Line 386: delete first ‘the’ (children); Line 391: ‘concordance’; Line 397: ‘hands’; Line 398: ‘wash their hands’; Line 400: ‘geohelminthiasis’; Line 403: sanitation ‘facilities’; Line 408: delete ‘were’; Line 409: 15% were positive ‘for’ T. trichura; Line 419: delete ‘periodically’; Line 431: delete ‘”’; Line 432: ‘practiced by the villagers’; Line 433: delete first ‘the’; Line 439: replace ‘of’ with ‘to their’; Lines 439-440: ‘trained at a very young age on the use of the toilet and wearing footwear outside the house’

Acknowledgements: Line 449: replace ‘which’ with ‘wish’; Line 450: for assisting in ‘conducting’ this research successfully. delete ‘conducted’.

Reviewer #2: The authors have discussed the Prevalence and risk factors of geohelminthiasis among the rural village children in Kota Marudu, Sabah, Malaysia. Of the 13 villages surveyed, geohelminthiasis was found prevalent only in 7, while there was no infection reported among the subjects of 6 villages. Though the authors have carried out extensive statistical analyses of data, the sample size in the study seems rather small. The results obtained through this study appear to be a corroboration of earlier studies carried out elsewhere across the globe.

The introductory section on geohelminthiasis has been reduced from 3 to 1 paragraph.

Parts of the discussion have been revised.

[Eg., Prevalence and risk factors of soil-transmitted helminthiasis ...

bmcpublichealth.biomedcentral.com › articles;

Prevalence & risk factors for soil transmitted helminth infection ...

www.ncbi.nlm.nih.gov › pmc › articles › PMC3994744;

Prevalence and risk factors associated with the presence of ...

www.ncbi.nlm.nih.gov › pmc › articles › PMC4554591;

Prevalence & risk factors for soil transmitted helminth ...

www.researchgate.net › publication › 260610263_Prevalence_risk_fact.;

Prevalence and risk factors associated with worm infestation ...

www.researchgate.net › publication › 4181077_Prevalence_and_risk_f...;

Geohelminth Infections and Nutritional Status of Preschool ...

www.hindawi.com › journals › scientifica;

Prevalence, Intensity of Soil-Transmitted Helminths, and ...

www.hindawi.com › journals › jtm;

The prevalence, intensities and risk factors associated with ...

onlinelibrary.wiley.com › doi › full;

prevalence and risk factors associated with worm infestation in ...

www.bioline.org.br › pdf- to name a few].

I wish the authors had highlighted the novel findings emerging from their study and avoided a lengthy textbook account in ‘Introduction’ and generalizations in ‘Discussion’.

---

## [Decision Letter · Decision Letter 1]

7 Sep 2020

PONE-D-20-17984R1

Prevalence and risk factors of geohelminthiasis among the rural village children in Kota Marudu, Sabah, Malaysia

PLOS ONE

Dear Dr. Chua,

Thank you for submitting your manuscript to PLOS ONE. After careful consideration, we feel that it has merit but does not fully meet PLOS ONE’s publication criteria as it currently stands. Therefore, we invite you to submit a revised version of the manuscript that addresses the points raised during the review process.

Thank you very much for revising your article, which looks much improved than before. Please find a minor suggestion from one of the reviewers, which I would again like to put for your kind consideration.

- The last two paras of conclusion may be reworked. 

Herein, I feel you may consider highlights of main findings/recommendations, may be in a single para. The last para of abstract may give you some idea about highlights/recommendations of study.

We look forward to receiving your revised manuscript.

Kind regards,

Arun K Yadav, Ph.D.

Academic Editor

PLOS ONE

Additional Editor Comments (if provided):

Thank you very much for revising your article, which looks much improved than before. Please find a minor suggestion from one of the reviewers, which I would again like to put for your kind consideration.

- The last two paras of conclusion may be reworked.

Herein, I feel you may consider highlights of main findings/recommendations, may be in a single para. The last para of abstract may give you some idea about highlights/recommendations of study.

Reviewers' comments:

Reviewer's Responses to Questions

**Comments to the Author**

1. If the authors have adequately addressed your comments raised in a previous round of review and you feel that this manuscript is now acceptable for publication, you may indicate that here to bypass the “Comments to the Author” section, enter your conflict of interest statement in the “Confidential to Editor” section, and submit your "Accept" recommendation.

Reviewer #1: All comments have been addressed

Reviewer #2: All comments have been addressed

2. Is the manuscript technically sound, and do the data support the conclusions?

Reviewer #1: (No Response)

Reviewer #2: Yes

3. Has the statistical analysis been performed appropriately and rigorously? 

Reviewer #1: (No Response)

Reviewer #2: Yes

4. Have the authors made all data underlying the findings in their manuscript fully available?

Reviewer #1: (No Response)

Reviewer #2: Yes

5. Is the manuscript presented in an intelligible fashion and written in standard English?

Reviewer #1: (No Response)

Reviewer #2: Yes

6. Review Comments to the Author

Reviewer #1: (No Response)

Reviewer #2: My concerns raised in the earlier review have been addressed in the revised manuscript . However, a minor revision is still required. In Discussion' section- the last two paragraphs are actually 'conclusion' and/ or recommendations meant for policy makers and social/administrative bodies. These paras may be reworked so as to avoid generalized statements and give crisp/ abridged recommendations in a couple of sentences.

7. PLOS authors have the option to publish the peer review history of their article (what does this mean?). If published, this will include your full peer review and any attached files.

Reviewer #1: No

Reviewer #2: No

---

## [Author Response · Author response to Decision Letter 1]

9 Sep 2020

Reviewer #2: My concerns raised in the earlier review have been addressed in the revised manuscript . However, a minor revision is still required. In Discussion' section- the last two paragraphs are actually 'conclusion' and/ or recommendations meant for policy makers and social/administrative bodies. These paras may be reworked so as to avoid generalized statements and give crisp/ abridged recommendations in a couple of sentences.

The 2 paragraphs have now to been reduced to one shown below:

In conclusion, this study clearly shows that geohelminthiasis still occurs with an overall prevalence of 14.3% among children of rural communities of Kota Marudu in Sabah. Infection by Ascaris had the highest rate at 67% compared to hookworm or T. trichiura. The most practical means of controlling geohelminthiasis in these communities would be expanding mass deworming programmes for school-age children, improving sanitation and water supplies, upgrading the socio-economic status and continually conducting health education programmes. The data on the prevalence of geohelminthiasis in this study would contribute to better public health monitoring and operation to reduce the infection in rural areas.

---

## [Editor Report · Decision Letter 2]

11 Sep 2020

Prevalence and risk factors of geohelminthiasis among the rural village children in Kota Marudu, Sabah, Malaysia

PONE-D-20-17984R2

Dear Dr. Chua,

We’re pleased to inform you that your manuscript has been judged scientifically suitable for publication and will be formally accepted for publication once it meets all outstanding technical requirements.

Kind regards,

Arun K Yadav, Ph.D.

Academic Editor

PLOS ONE

Additional Editor Comments (optional):

Thank you very much for your sincere revision. The manuscript now appears suitable for publication.
---

## [Editor Report · Acceptance letter]

16 Sep 2020

PONE-D-20-17984R2

Prevalence and risk factors of geohelminthiasis among the rural village children in Kota Marudu, Sabah, Malaysia

Dear Dr. Chua:

I'm pleased to inform you that your manuscript has been deemed suitable for publication in PLOS ONE. Congratulations! Your manuscript is now with our production department.

Kind regards,

on behalf of

Dr. Arun K Yadav 

Academic Editor

PLOS ONE